# Toward a Unified Geometry Understanding: Riemannian Diffusion Framework for Graph Generation and Prediction

**Yisen Gao**[1,2,4]**, Xingcheng Fu**[1]***, Qingyun Sun**[3,4]**, Jianxin Li**[4]**, Xianxian Li**[1]

[1]Key Lab of Education Blockchain and Intelligent Technology, Guangxi Normal University
[2]Computer Science and Engineering, The Hong Kong University of Science and Technology
[3]Guangxi Key Lab of Multi-source Information Mining & Security, Guangxi Normal University
[4]School of Computer Science and Engineering, Beihang University
ygaodi@cse.ust.hk, {fuxc, lixx}@gxnu.edu.cn, {sunqy,lijx}@buaa.edu.cn

## Abstract

Graph diffusion models have made significant progress in learning structured graph data and have demonstrated strong potential for predictive tasks. Existing approaches typically embed node, edge, and graph-level features into a unified latent space, modeling prediction tasks including classification and regression as a form of conditional generation. However, due to the non-Euclidean nature of graph data, features of different curvatures are entangled in the same latent space without releasing their geometric potential. To address this issue, we aim to construt an ideal Riemannian diffusion model to capture distinct manifold signatures of complex graph data and learn their distribution. This goal faces two challenges: numerical instability caused by exponential mapping during the encoding proces and manifold deviation during diffusion generation. To address these challenges, we propose **GeoMancer**: a novel Riemannian graph diffusion framework for both generation and prediction tasks. To mitigate numerical instability, we replace exponential mapping with an isometric-invariant Riemannian gyrokernel approach and decouple multi-level features onto their respective task-specific manifolds to learn optimal representations. To address manifold deviation, we introduce a manifold-constrained diffusion method and a self-guided strategy for unconditional generation, ensuring that the generated data remains aligned with the manifold signature. Extensive experiments validate the effectiveness of our approach, demonstrating superior performance across a variety of tasks.

## 1 Introduction

Graph-structured data is widely used in real-world applications [2] such as recommendation systems [3], social networks [4], and molecular modeling [5]. Due to the non-Euclidean structure of graphs, there are some studies [6] that have leveraged differential geometry to explore non-Euclidean geometric spaces that better align with the intrinsic structure of graphs. In a geometric perspective, non-Euclidean manifolds can enable a deeper understanding of graph-structured data, which in turn facilitates improved performance in downstream tasks. For example, hyperbolic spaces are well-suited for modeling small-world and hierarchical graphs [7, 8], whereas spherical spaces are more effective at capturing the structure of densely connected graphs [9]. Product manifold [10, 11] is introduced to model more complex graph data by constructing a space of mixed curvatures [12].

---

*Corresponding author

39th Conference on Neural Information Processing Systems (NeurIPS 2025).

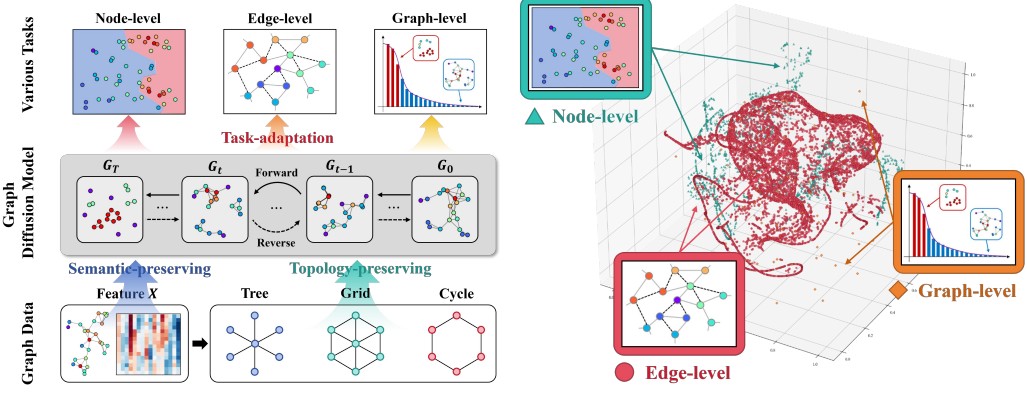

(a) Overview of Graph Diffusion Model     (b) Complex Multi-level Data Manifolds

Figure 1: (a) An overview of latent graph diffusion model for prediction and generation. (b) Feature Entanglements: we visualize the multi-level latent representations learned in graph regression tasks on the ZINC12K [1] dataset using t-SNE. The results reveal that representations with distinct geometric properties become entangled in a shared Euclidean latent space.

Moreover, diffusion models [13] have demonstrated strong capabilities in capturing complex data distributions and understanding the inherent geometric properties of the data [14]. Recent methods [15] have introduced the latent graph diffusion framework for unifying the graph generation and prediction tasks. Specifically, it embeds multi-level graph features (including node-level, edge-level, and graph-level) into a unified low-dimensional latent space. Then the prediction task is reformulated as a conditional generation problem, where graph features explicitly serve as the condition guiding the generation of target graph attributes/labels.

However, although this latent diffusion framework is theoretically sound in general data, it overlooks the rich and diverse geometric structures inherently present in graph data and lacks a unified geometric perspective for graph learning tasks. As observed in Fig. 1, representations across different levels are often entangled within a shared latent space, despite exhibiting distinct intrinsic geometric properties. These features exhibit curvature heterogeneity across different levels and should be better modeled in spaces with varying curvature. This highlights the limitations of the existing method and calls for a new modeling paradigm capable of capturing the optimal manifold signatures [16] (such as the choice of curvature, manifold components, and dimensionality) for complex graph data.

Based on the above insights, an ideal Riemannian diffusion framework should first reconstruct the underlying data manifold using a geometry-aware autoencoder, and then model the distribution over this manifold through a diffusion process. However, the inherent geometric complexity of multi-level graph features, further compounded by the demands of multi-task learning, makes modeling the underlying manifold highly challenging and well beyond the capacity of simple designs. Specifically, it faces the following two key challenges:

*How to assign an appropriate manifold signature during the autoencoding process?* A common approach is to map features on product manifolds [10, 17] with learnable curvatures [18] using exponential mapping. However, due to curvature heterogeneity across different feature levels, this method may lead to numerical instability in the exponential map, making model optimization more difficult and restricting its effectiveness on a range of downstream tasks.

*How to generate an accurate manifold distribution during the diffusion process?* In the generation stage, diffusion models often deviate from the original data manifold [19], leading to sub-optimal performance. One approach [20] to address this issue is by incorporating manifold constraints into the condition generation process, which helps guide the model back to the desired manifold. This can be seen as a form of precise conditional control. However, in unconditional graph generation tasks, the absence of such guiding information can result in deviations from the intended manifold structure.

To address these challenges, we propose **GeoMancer**: a novel Riemannian Diffusion Framework for graph generation and prediction. To better choose the manifold signatures, we construct a product manifold as the latent space for each level feature. Then, we decouple the multi-level features entangled within it onto their corresponding task-specific manifolds. Additionally, to mitigate numerical

instability caused by exponential and logarithmic mappings, we employ a Riemannian kernel method based on generalized Fourier transforms. This approach preserves the isometric geometric properties of Riemannian spaces while remaining compatible with well-established Euclidean models. To guide the diffusion model to generate features on a more desirable manifold, we leverage the rich geometric information in the latent space to produce pseudo-labels for unconditional graph generation. This allows us to reformulate all graph-related tasks as conditional generation problems. During the generation process, we further enhance this guidance by introducing a manifold-constrained sampling strategy. Our contributions are summarized as follows:

- We propose a unified Riemannian diffusion framework that provides a geometric understanding of graph generation and prediction tasks by assigning geometric spaces aligned with the intrinsic property of multi-level graph data.
- We introduce key improvements to both the encoding and generation stages of the Riemannian latent diffusion framework, including multi-level product manifold modeling, a numerically stable Riemannian kernel method, and a manifold-constrained conditional generation strategy, enabling more accurate and robust learning of the underlying geometry in graph data.
- Extensive experiments demonstrate that our model achieves excellent performance across multiple levels of tasks in generation, classification, and regression.

## 2   Related Work

**Graph diffusion model for generation.** Graph diffusion models can be broadly categorized into two approaches: discrete diffusion and latent diffusion. In discrete diffusion, GDSS [21] employs stochastic differential equation (SDE)-based diffusion techniques to model both node features and adjacency matrices. GSDM [22] extends this framework by incorporating diffusion in the spectral domain, further enhancing its capability to capture graph structures. DiGress [23] adapts the diffusion process specifically for discrete data, while GruM [24] introduces a Schrödinger bridge to preserve the effective structural properties of graphs during generation. Defog [25] proposes a discrete flow matching method for generating discrete graph-structured data. In latent diffusion, Graphusion [26] utilizes variational autoencoders to map graph structures into a latent representation space, assigning soft labels through structural clustering to capture spatial relationships. HypDiff [27] projects graph structures into hyperbolic space and applies geometrically constrained diffusion to maintain the anisotropic properties of graphs, ensuring that the generated structures align with their intrinsic geometric characteristics.

**Graph diffusion model for representation.** Compared to their widespread use in generative tasks, graph diffusion models have been relatively under-explored in the context of representation learning. Among the few existing approaches, DDM [28] focuses on denoising node features and leverages a Unet-based architecture to extract effective representations. Meanwhile, LGD [15] represents a significant advancement as the first model to unify generation and diffusion within a single framework. It achieves representation learning by reformulating downstream tasks as conditional diffusion processes, thereby bridging the gap between generative and discriminative objectives.

**Riemannian representation model.** Representation learning in Riemannian space primarily relies on exponential and logarithmic mappings. For example, HGCN [7] uses exponential mapping to project representations generated by GCN into hyperbolic space, performing operations like aggregation and activation in Euclidean space after logarithmic mapping. [12] extends this to Riemannian spaces with multiple curvatures, enabling more flexible representation learning. HyLA [29] introduces a hyperbolic framework that replaces exponential mappings with isometric invariant kernel mappings, preserving hyperbolic geometric properties. Building on this, MotifRGC [18] generalizes the approach to arbitrary Riemannian spaces and uses contrastive learning to assign node-specific curvatures, significantly improving representation expressiveness. The Riemannian MOE architecture [30, 31] has also been introduced to capture different graph structure features recently.

## 3   Method

In this section, we propose our model GeoMancer, a Riemannian diffusion model for graph generation and prediction. In Section 3.1, we first introduce how to use a graph diffusion model as a unified framework for both generation and prediction tasks. In Section 3.2, we propose the Riemannian graph

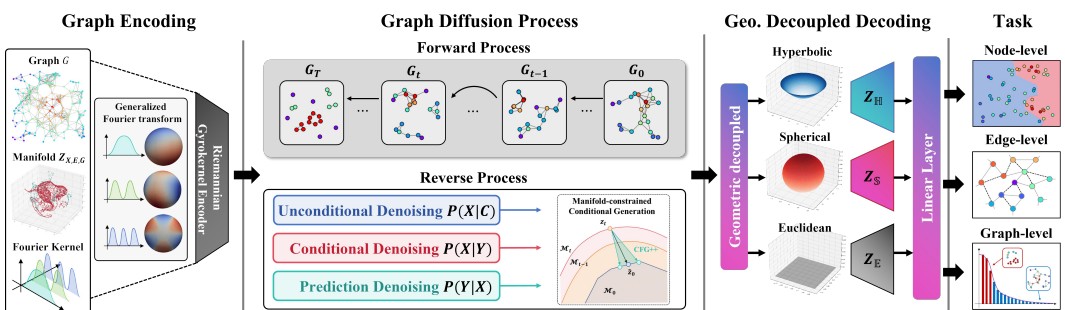

Figure 2: An Illustration of GeoMancer Architecture.

autoencoder. We first introduce a Riemannian kernel based on the generalized Fourier transform to replace the exponential map. Then, we describe how the encoder captures the complex geometry of the latent space, and how the decoder projects data onto the task-specific manifold. In Section 3.3, we introduce a self-guided strategy for unconditional graph generation, allowing all graph tasks to be unified under a conditional generation framework. We further incorporate a manifold constraint guidance method to ensure the generation aligns with clean data manifolds. The preliminaries for the methods are presented in the Appendix B.

## 3.1 Notation and Problem Definition

A graph with $N$ nodes is defined as $G = (X, E, Y_X, Y_G)$, where $X = [x_1, x_2, \ldots, x_N] \in \mathbb{R}^{N \times d_n}$ denotes the node features and $x_i$ denotes the feature of node $i$. $E \in \mathbb{R}^{N \times N \times d_e}$ denotes the feature of edges and $e_{ij}$ is the edge feature between node $i$ and node $j$. $Y_X \in \mathbb{R}^{N \times 1}$ denotes labels or properties of the nodes, $Y_G \in \mathbb{R}$ denotes the label or property of the graph $G$.

Our model is a generative framework that unifies general generation and multi-level prediction tasks under a diffusion paradigm. Specifically, it can be conceptualized as follows: (1) For the unconditional graph generation task, the objective is to learn a graph generative model that can capture the distribution $p(G)$ of the target graph set $\{G_1, G_2, \ldots, G_N\}$. (2) For the conditional graph generation task, it seeks to generate synthetic graphs conditioned on specific label or property $Y_G$. The goal is to model the conditional distribution $p(G|Y_G)$. (3) For the downstream prediction task, it can be viewed as a special form of the conditional generation task where the generation target is label $Y$ and the condition is $(X, E)$. Thus, the generation model is to learn the conditional distribution of $p(Y|X, E)$.

## 3.2 Riemannian GyroKernel Autoencoder

The goal of Riemannian autoencoder is to encode complex multi-level graph features into a unified low-dimensional latent space that preserves the geometric heterogeneity and decode them to the task-specific manifold. However, mapping data onto a product manifold often relies on exponential and logarithmic maps, which are prone to severe numerical instability, making the optimization difficult.

To address this, we use a Riemannian kernel [18] as a substitute, which maps Riemannian prior representations to Euclidean space while preserving isometry for encoding Euclidean features. Guided by Bochner's Theorem (See the Appendix A), isometry-invariant kernels can be constructed through Fourier mappings based on eigenfunctions of Laplace operators. In this work, we utilize a generalized Fourier mapping $\phi_{\mathrm{gF}}(x)$ defined in the gyrovector ball $\mathbb{G}_\kappa^n$ of Riemannian manifold with the learnable curvature $\kappa$. Specifically, the eigenfunction $\mathrm{gF}_{\boldsymbol{\omega}, b, \lambda}^\kappa(\boldsymbol{x})$ in the gyrovector ball $\mathbb{G}_\kappa^n$ can be formulated as:

$$\mathrm{gF}_{\boldsymbol{\omega}, b, \lambda}^\kappa(\boldsymbol{x}) = A_{\boldsymbol{\omega}, \boldsymbol{x}} \cos\left(\lambda \langle \boldsymbol{\omega}, \boldsymbol{x} \rangle_\kappa + b\right), \boldsymbol{x} \in \mathbb{G}_\kappa^n, \tag{1}$$

where $A_{\boldsymbol{\omega},\boldsymbol{x}} = \exp\left(\frac{n-1}{2}\langle\omega, x\rangle_\kappa\right)$. $\langle\boldsymbol{\omega},\boldsymbol{x}\rangle_\kappa = \log\frac{1+\kappa\|\boldsymbol{x}\|^2}{\|\boldsymbol{x}-\boldsymbol{\omega}\|^2}$ is the signed distance in the gyrovector ball. $\boldsymbol{\omega}$ denotes the phase vector, uniformly sampling from a a n-dimensional unit ball. $b$ denotes the bias, uniformly sampling from $[0, 2\pi]$.

Then the generalized Fourier mapping $\phi_{\mathrm{gF}}(x)$ can be denoted as:

$$\phi_{\mathrm{gF}}(x) = \frac{1}{\sqrt{m}}\left[\mathrm{gF}^\kappa_{\boldsymbol{\omega}_1,\lambda_1,b_1}(x),\cdots,\mathrm{gF}^\kappa_{\boldsymbol{\omega}_m,\lambda_m,b_m}(x)\right] \in \mathbb{R}^m. \tag{2}$$

We initialize a product manifold Riemannian representation vectors $V_\kappa = \{V_{\kappa_1}, V_{\kappa_2}, \ldots, V_{\kappa_m}\}$ with different curvatures $\kappa_i$. Using equation (2), we compute the isometric-invariant Euclidean features $\overline{V}_\kappa$ after generalized Fourier mapping:

$$\overline{V}_\kappa = \phi_{\mathrm{gF}}(V_\kappa) = \{\phi_{\mathrm{gF}}(V_{\kappa_1}), \phi_{\mathrm{gF}}(V_{\kappa_2}), \ldots, \phi_{\mathrm{gF}}(V_{\kappa_m})\}. \tag{3}$$

Then, we can aggregate each dimension of any graph representation $Z$ by taking advantage of the geometric properties of the product manifold: $\overline{Z} = Z\overline{V}_\kappa$. Unlike simple feature mapping, this approach enables each dimension of the features to capture distinct geometric information, thereby more effectively revealing the intrinsic geometric properties of graph features. We have provided a more detailed introduction to this method in Appendix B.

**Encoder**. Here, we need to build a powerful graph encoder that can embed the node features $X$ and edge features $E$ of the graph $G$ into a unified low-dimension latent space $Z = \{Z_X, Z_E\} \in \mathbb{R}^{(N+N\times N)\times d}$. The graph-level feature $Z_G$ can be obtained by aggregating $Z_X$ and $Z_E$. To better represent edge features $E$, we adopt a flexible graph transformer [32] as the backbone network for edge enhancement. It constructs relevant attention mechanisms for both nodes $Z_{x_i}$ and edges $\boldsymbol{z}_{e_{ij}}$ to efficiently underlying the relationships between them. Specifically, the $l$-th graph transformer layer can be represented as:

$$\begin{aligned}
\boldsymbol{z}^{l+1}_{e_{ij}} &= \sigma\left(\rho\left((\boldsymbol{Q}\boldsymbol{z}^l_{x_i}, \boldsymbol{K}\boldsymbol{z}^l_{x_j}) \odot \boldsymbol{E}_w\boldsymbol{z}^l_{e_{ij}}\right) + \boldsymbol{E}_b\boldsymbol{z}^l_{e_{ij}}\right), \\
\alpha_{ij} &= \mathrm{Softmax}_{j\in\mathcal{V}}(\boldsymbol{W}\boldsymbol{z}^{l+1}_{e_{ij}}), \\
\boldsymbol{z}^{l+1}_{x_i} &= \sum_{j\in\mathcal{V}} \alpha_{ij}(\boldsymbol{V}\boldsymbol{z}^l_{x_j} + \boldsymbol{E}_v\boldsymbol{z}^{l+1}_{e_{ij}}),
\end{aligned} \tag{4}$$

where $\boldsymbol{Q}, \boldsymbol{K}, \boldsymbol{V}, \boldsymbol{W}, \boldsymbol{E}_w, \boldsymbol{E}_b, \boldsymbol{E}_v$ are learnable weight matrices; $\odot$ denotes the elementwise multiplication; $\sigma$ is a nonlinear activation and $\rho(x) = (\mathrm{ReLU}(\mathbf{x}))^{1/2} - (\mathrm{ReLU}(-\mathbf{x}))^{1/2}$ is a function used for training stability.

Then, the geometric latent representation $\overline{Z}$ is obtained by endowing the embeddings with product manifold geometric properties with a given $\overline{V}_\kappa$. To simplify notation, we denote the geometric latent representation $\overline{Z}$ as $Z$ in the remainder.

**Decoder.** The decoder aims to project the latent representations $Z$ onto task-specific manifolds, enabling effective adaptation to downstream tasks. To achieve this, we design a dedicated decoupling modle for each level feature. The core idea of this method is to split a complex product manifold into multiple simple manifolds: $\mathcal{M} \to \mathcal{M}_1 \times \cdots \times \mathcal{M}_m$ and selects the most appropriate geometric representation based on the specific requirements of each task.

**Proposition 3.1.** *Let $f_i : \mathcal{M}_i \to \mathbb{R}, i \in \{1, 2, \ldots, L\}$ be twice-differentiable functions such that $\Delta_{\mathcal{M}_i}f_i = \lambda_i f_i$, where $\Delta_{\mathcal{M}_i}$ is Laplace operators and $\lambda_i$ is the eigenvalue. Take $\pi_i : \mathcal{M} \to \mathcal{M}_i$ to be the projection of $M$ onto $M_i$, We can then define the natural extension of $f_i$ to $M$ via $g_i = f_i \circ \pi_i$. It follows that*

$$\Delta_{\mathcal{M}}\left(\prod_{i=1}^L g_i\right) = \left(\sum_{i=1}^L \lambda_i\right)\prod_{i=1}^L g_i. \tag{5}$$

According to the proposition 3.1, a Euclidean representation endowed with the geometric prior of a complex product manifold can be decoupled into simpler representations over its constituent manifolds.

Then, we decompose $Z = [Z_{\kappa_1}, Z_{\kappa_2}, \ldots, Z_{\kappa_m}]$ into the representation of each component $Z_{\kappa_i}$. Since they are still in Euclidean space after the generalized Fourier mapping, we directly use the attention or linear layers at the end to capture the most effective manifold representation for the task.

**Training Objective**. The training objective encompasses a target loss $\mathcal{L}_{tgt}$ and a regularization constraints $\mathcal{L}_{reg}$:

$$\mathcal{L} = \mathcal{L}_{tgt} + \mathcal{L}_{reg}. \tag{6}$$

In the generation task, the target loss $\mathcal{L}_{tgt}$ is composed of the reconstruction cross-entropy loss of nodes and edges. For regression or classification tasks, the target loss $\mathcal{L}_{tgt}$ is the MSE of a regression task or the cross-entropy of a classification task. The regularization loss $\mathcal{L}_{reg}$ represents the regularization constraints, which pushes the representation $Z$ towards a standard normal distribution, thus preventing high variance in the latent space. Specifically, it can be written as: $L_{reg} = D_{\text{KL}}\left(q(Z \mid (X, E)) \parallel N(0, I)\right)$.

## 3.3 Manifold-Constrained Diffusion

After obtaining a geometric latent representation $Z = \{Z_X, Z_E, Z_G\}$, we conduct diffusion training and generation within this unified latent space. In addition to incorporating geometric priors, a notable advantage of the Riemannian kernel method is that the representations remain in Euclidean space. This allows for the direct application of classical diffusion techniques without the design for more complex Riemannian diffusion. The forward process [13] of the diffusion model gradually turns the data toward noise. Specifically, this process can be defined as:

$$q(Z_t|Z_{t-1}) = N(X_t; \sqrt{\bar{\alpha}_{t-1}}Z_{t-1}, \sqrt{1 - \bar{\alpha}_{t-1}}I) \tag{7}$$

where $N(\cdot)$ is the Gaussian distribution and $\bar{\alpha}_{t-1}$ is calculated by noise schedule.

To better capture the complex data manifolds, we hope to adopt a manifold-constrained conditional generation approach [20]. While tasks like conditional generation and prediction can be naturally redefined as conditional generation problems, unconditional graph generation lacks explicit label guidance. To bridge this gap, we introduce a self-guided mechanism that leverages the rich geometric information embedded in the latent space to generate pseudo-labels, thereby providing effective guidance to the model during generation.

**Self-Guidance**. Even in unconditional generation, structural variations inherently reflect differences on the underlying geometric manifold. Therefore, a natural approach is to leverage the rich geometric features encoded in the latent space to guide the graph generation process. By applying k-means clustering to the latent graph-level representation $Z_G$, we assign a pseudo-label $C$ to each graph. Consequently, the unconditional generation of graphs can be reformulated as a new conditional generation task to learn $P(G|C)$. In the sampling stage, $C$ is selected randomly for each graph.

Further, we unify graph generation and prediction task by modeling them within a general conditional generation framework to learn the distribution $P(x|y)$. For unconditional generation, we generate pseudo labels $C$ that capture complex geometric property through self-guidance. In conditional graph generation, explicit graph properties serve as conditions. Predictive tasks, such as classification and regression, are reformulated as conditional generation processes based on known representations.

We then adopt CFG++ [20], a manifold-constrained classifier-free guidance method that formulates conditional generation as an inverse problem, enabling the model to better capture the clean manifold of the data. Specifically, the reverse generation process can be formulated as:

$$\begin{aligned}
\tilde{Z}_0 &= (Z_t - \sqrt{1 - \bar{\alpha}_t}\tilde{\epsilon}_\theta(Z_t, \tau(y)))/\sqrt{\bar{\alpha}_t} \\
Z_{t-1} &= \sqrt{\bar{\alpha}_{t-1}}\tilde{Z}_0 + \sqrt{1 - \bar{\alpha}_{t-1}}\epsilon_\theta(Z_t)
\end{aligned} \tag{8}$$

where $y$ is the condition and $\tau(y)$ is the latent embedding of $y$. $\tilde{\epsilon}_\theta$ is derived from the outputs of the model $\epsilon_\theta$ under both conditional and unconditional settings. Specifically, it can be written as:

$$\tilde{\epsilon}_\theta(\hat{Z}_t, \tau(\boldsymbol{y})) = (1 - \lambda)\boldsymbol{\epsilon}_\theta(\hat{Z}_t, \tau(\boldsymbol{y})) - \lambda\boldsymbol{\epsilon}_\theta(\hat{Z}_t) \tag{9}$$

where $\lambda$ is a hyperparameter that controls the strength of condition guidance.

The model $\epsilon_\theta$ is optimized via a noise prediction strategy, learning to estimate the added noise in the forward process:

$$\mathcal{L}_{diff} = \mathbb{E}_{Z_t, \boldsymbol{y}, \epsilon_t \sim \mathcal{N}(\mathbf{0}, \mathbf{I}), t} \left[\|\tilde{\epsilon}_\theta(Z_t, t, \tau(\boldsymbol{y}) - \epsilon_t\|_2^2\right], \tag{10}$$

Table 1: Unconditional generation results on QM9. (**Bold**: best; Underline: runner-up.)

| Model | Validity (%)↑ | Uniqueness (%)↑ | FCD↓ | NSPDK↓ | Novelty (%)↑ |
|---|---|---|---|---|---|
| MoFlow | 91.36 | 98.65 | 4.47 | 0.0170 | 94.72 |
| GraphAF | 74.43 | 88.64 | 5.27 | 0.0200 | 86.59 |
| GraphDF | 93.88 | **98.58** | 10.93 | 0.0640 | **98.54** |
| GDSS | 95.72 | 98.46 | 2.90 | 0.0003 | 86.27 |
| DiGress | 99.01 | 96.34 | 0.25 | 0.0003 | 35.46 |
| HGGT | 99.22 | 95.65 | 0.40 | 0.0003 | 24.01 |
| GruM | 99.69 | 96.90 | 0.11 | **0.0002** | 24.15 |
| LGD | 98.46 | 97.53 | 0.32 | 0.0004 | 56.35 |
| GeoMancer | **100.00** | 95.74 | **0.09** | **0.0002** | 90.43 |

Table 2: Conditioal generation results on QM9 (MAE ↓). (**Bold**: best; Underline: runner-up.)

| Method | $\mu$ | $\alpha$ | $\epsilon_{\text{HOMO}}$ | $\epsilon_{\text{LUMO}}$ | $\Delta\epsilon$ | $C_V$ |
|---|---|---|---|---|---|---|
| $\omega$ | 0.043 | 0.10 | 39 | 36 | 64 | 0.040 |
| $\omega$ (LGD) | 0.058 | 0.06 | 18 | 24 | 28 | 0.038 |
| $\omega$ (GeoMancer) | 0.054 | 0.06 | 16 | 22 | 27 | 0.037 |
| Random | 1.616 | 9.01 | 645 | 1457 | 1470 | 6.857 |
| Natom | 1.053 | 3.86 | 426 | 813 | 866 | 1.971 |
| EDM | 1.111 | 2.76 | 356 | 584 | 655 | 1.101 |
| GeoLDM | 1.108 | **2.37** | 340 | **522** | 587 | 1.025 |
| LGD | 0.879 | 2.43 | 313 | 641 | 586 | **1.002** |
| GeoMancer | **0.832** | 2.38 | **304** | 628 | **581** | 1.002 |

**Model Architechture**. Since there is no prior information about edges during noise sampling, we do not use message passing or positional encoding. Instead, we directly employ a graph transformer as the denoising network. For incorporating conditional information, we follow existing methods [15] and introduce the condition $\boldsymbol{y}$ through the cross-attention:

$$
\begin{aligned}
\boldsymbol{z}_{x_i}^{l+1} &= \text{softmax}\left(\frac{(\boldsymbol{Q}_h \boldsymbol{z}_{x_i}^l)(\boldsymbol{K}_h \tau(\boldsymbol{y}))^\top}{\sqrt{d'}}\right) \cdot \boldsymbol{V}_h \tau(\boldsymbol{y}), \\
\boldsymbol{z}_{e_{ij}}^{l+1} &= \text{softmax}\left(\frac{(\boldsymbol{Q}_e \boldsymbol{z}_{e_{ij}}^l)(\boldsymbol{K}_e \tau(\boldsymbol{y}))^\top}{\sqrt{d'}}\right) \cdot \boldsymbol{V}_e \tau(\boldsymbol{y}).
\end{aligned}
\tag{11}
$$

## 4 Experiment

### 4.1 Experimental Setup

We evaluate the effectiveness of our model[2] by conducting experiments across multiple tasks. For graph structure generation, we assess both unconditional and conditional molecular generation. For prediction tasks, we evaluate the model's performance on node classification and graph regression. These tasks collectively cover node-level, edge-level and graph-level tasks, providing a thorough assessment of our model's capabilities.

For all baselines, we report their results based on the results presented in their papers or the optimal parameters provided. All experiments were conducted using PyG, and the reported results are averaged over five runs. All models were trained and evaluated on an Nvidia A800 80GB GPU. More experimental details are reported in Appendix C.

### 4.2 Generation Task

We conduct molecular graph generation experiments on the QM9 dataset [33], a widely used benchmark in machine learning for molecular data. QM9 contains 133,885 molecular graphs with 12

---

[2]Our code is available at https://github.com/RingBDStack/GeoMancer.

Table 3: Node-level classification tasks (accuracy ↑) ( **Bold**: best; Underline: runner-up. OOM: cuda out of memory.)

| Model | Photo | Physics | Pubmed | Cora | Citeseer |
|---|---|---|---|---|---|
| GCN | 92.70 ± 0.20 | 96.18 ± 0.07 | 88.9 ± 0.32 | 81.60 ± 0.40 | 71.60 ± 0.40 |
| GAT | 93.87 ± 0.11 | 96.17 ± 0.08 | 83.28 ± 0.12 | 83.00 ± 0.70 | 72.10 ± 1.10 |
| GraphSAINT | 91.72 ± 0.13 | 96.43 ± 0.05 5 | 85.64 ± 0.26 | 81.82 ± 0.22 | 72.30 ± 0.17 |
| Graphormer | 92.74 ± 0.13 | OOM | 92.64 ± 0.96 | 82.62 ± 0.12 | 71.60 ± 0.32 |
| GraphGPS | 95.06 ± 0.13 | OOM | 90.28 ± 0.62 | 82.84 ± 0.13 | **72.73 ± 0.23** |
| Exphormer | 95.35 ± 0.22 | 96.89 ± 0.09 | 91.44 ± 0.59 | 82.77 ± 0.38 | 71.63 ± 0.29 |
| NAGphormer | 95.49 ± 0.11 | 97.34 ± 0.03 | 91.76 ± 0.49 | 82.13 ± 1.18 | 71.40 ± 0.30 |
| LGD | 96.94 ± 0.14 | 98.55 ± 0.12 | 92.88 ± 0.29 | 82.81 ± 1.18 | 72.40 ± 0.30 |
| GeoMancer | **97.05 ± 0.13** | **98.78 ± 0.12** | **93.10 ± 0.29** | **83.50 ± 0.23** | 72.60 ± 0.20 |

Table 4: Ablation study on unconditional generation. (**Bold**: best; Underline: runner-up.)

| Model | Validity (%)↑ | Uniqueness (%)↑ | FCD ↓ | NSPDK ↓ | Novelty (%)↑ |
|---|---|---|---|---|---|
| GeoMancer(w/o self-guidance) | 98.99 | **97.61** | 0.12 | 0.0003 | 54.06 |
| GeoMancer(w/o cfg++) | **100.00** | 91.74 | **0.09** | 0.0010 | 76.43 |
| GeoMancer(w/o Riemannian) | **100.00** | 92.53 | 0.25 | 0.0004 | 90.26 |
| GeoMancer | **100.00** | 95.74 | **0.09** | **0.0002** | **90.43** |

quantum chemical properties limited to 9 heavy atoms. **Unconditional Generation**. In the unconditional molecular generation task, we evaluate the model's ability to capture the distribution of molecular data and generate chemically valid and structurally diverse molecules. Specifically, validity is the fraction of valid molecules without valency correction or edge resampling. Uniqueness quantifies the proportion of unique valid molecules among the generated set. Novelty assesses the fraction of valid molecules that do not appear in the training set. To further evaluate the quality of the generated molecules, we employ two additional metrics: the Neighborhood Subgraph Pairwise Distance Kernel (NSPDK) MMD [34], which computes the Maximum Mean Discrepancy (MMD) between the generated and test molecules by considering both node and edge features, and the Fréchet ChemNet Distance (FCD) [35], which evaluates the distance between the training and generated graph sets using the activations of the penultimate layer of ChemNet, providing a measure of similarity in the feature space. For the baseline models, we selected the classical and recent state-of-the-art approaches, including MoFlow [36], GraphAF [37], GraphDF [38], GDSS [21], DiGress [23], HGGT [39], GruM [24] and LGD [15].

The results have been reported in Table 1. It can be observed that our model achieves significant improvements in the task of unconditional molecular generation. Specifically, we achieve state-of-the-art performance in terms of validity and the distributional similarity of molecular structures. In addition, our model shows competitive results in uniqueness and novelty. Notably, compared to LGD [15], which also employs a latent graph diffusion framework, our approach achieves a substantial improvement in novelty. We will further investigate the underlying reasons for this phenomenon through detailed ablation studies.

**Conditional Generation**. For the conditional generation task, its objective is to generate target molecules with specific chemical properties. Following the experimental setup outlined in [5], we split the training set into two halves, each containing 50,000 molecules. We train a latent graph diffusion model and a separate property prediction network on each subset. During evaluation, we generate a molecule using the latent graph diffusion model conditioned on a given property and then use the property prediction network to predict the target property $y$ for the generated molecule. We calculate the mean absolute error (MAE) between the predicted property and the true value, conducting experiments across six properties: Dipole moment $\mu$, polarizability $\alpha$, orbital energies $\epsilon_{HOMO}$, $\epsilon_{LUMO}$, their gap $\Delta\epsilon$ and heat capacity $C_V$. Following [15], we establish EDM [40], GeoLDM [5] and LGD [15] as baseline models. Additionally, we include the following reference points for comparison: (a) the MAE of the regression model $\omega$ of ours which serve as a lower bound of the generative models; (b) Random, which shuffle the labels and evaluate $\omega$, representing an upper

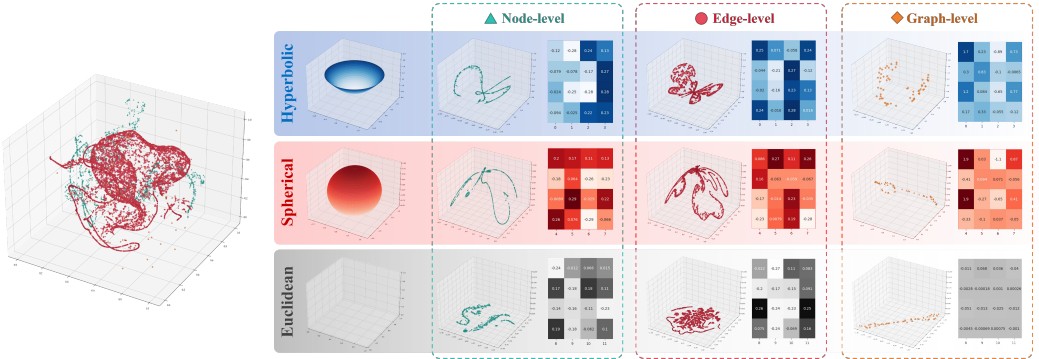

Figure 3: Visualization of the embeddings with different downstream task in the sub-spaces of the geometric decoupled decoders.

bound for the MAE metric; (c) Natoms, which predicts the properties based only on the number of atoms in the molecule. The results are in Table 2.

First, $\omega$ of our model achieves lower MAE compared to other models, demonstrating that the Riemannian autoencoder excels in regression capability. Additionally, our approach delivers outstanding performance across multiple properties in the generation task. Notably, even without incorporating 3D information, we successfully achieve high-quality conditional generation.

### 4.3 Prediction Task

For predictive tasks, we evaluated the graph regression task and node classification task.

**Graph Regression**. For graph regression task, we select ZINC12k [1], which is a subset of ZINC250k containing 12k molecules. The task focuses on predicting molecular properties, particularly constrained solubility, with performance evaluated by MAE. Here, we use the official split of the dataset. As baselines, we consider a range of representative graph regression models. GIN [41] employs a simple sum aggregator with learnable bias followed by MLP updates, achieving expressive power equivalent to the 1-WL test. PNA [42] combines multiple

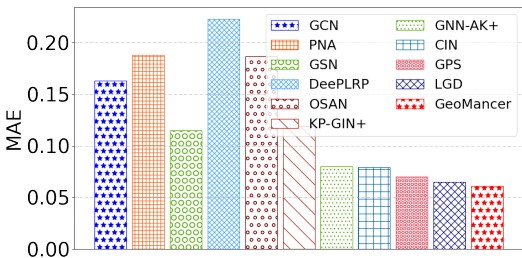

Figure 4: Comparison of graph regression task on Zinc12k dataset.

neighborhood aggregators with degree-scalers, allowing the model to capture diverse structural statistics. DeepLRP [43] encodes local structural patterns by pooling over permutations of nodes within small subgraphs. OSAN [44] introduces ordered subgraph sampling and aggregation to enhance message passing with subgraph-level information. KP-GIN+ [45] extends GIN with edge-sensitive updates while aggregating peripheral K-hop subgraphs for richer context. GNN-AK+ [46] improves expressiveness by applying a subgraph GNN to each node's local induced neighborhood. CIN [47] generalizes message passing beyond edges by propagating information across nodes, edges, and higher-order cells within a cell complex. GPS [48] integrates local MPNN aggregation with global Transformer attention in a hybrid design. Finally, LGD [15] applies latent-space diffusion to jointly model graph structure and multi-level features.

As shown in Fig. 4, our model demonstrates superior performance compared to traditional regression models. This fully demonstrates that GeoMancercan achieve better data understanding capabilities by effectively capturing the underlying geometric manifolds of complex multi-level data. The improvement over baselines highlights the model's ability to integrate both local and global geometric information in its representations. Furthermore, GeoMancerconsistently produces accurate predictions across different datasets and experimental settings, reflecting the stability of its learned embeddings. In addition, GeoMancerexhibits an excellent convergence rate, as described in Appendix D, making it efficient to train while maintaining high performance.

**Node Classification**. For node classification tasks, we evaluate our model on several widely used benchmark datasets [49]: Amazon Photo, a co-purchase network where nodes represent products and edges indicate frequent co-purchases; PubMed, a biomedical citation network; and Physics, a citation network from the arXiv Physics section;Cora, a citation network of scientific publications; Citeseer, a citation network of six research fields related to computer science. These datasets provide diverse scenarios for evaluating our model's ability to capture node-level semantics. The common 60%, 20%, 20% random split is adopted for the first three dataset, with the remaining adopting the standard split. We use Accuracy as the classification metric. Baselines include classic GNNs (GCN [50], GAT [51], GraphSAINT [52]) and graph transformers (Graphormer [53], SAN [54], GraphGPS [48], Exphormer [55], NAGphormer [56]).

The results are reported in Table 3. Our model achieves state-of-the-art performance on the node classification task, surpassing not only GNNs but also Graph Transformers. This demonstrates the model's ability to effectively capture the conditional probability distribution for regression tasks. Furthermore, our approach outperforms LGD, highlighting that the Riemannian diffusion mechanism successfully identifies the Riemannian manifold better suited for node classification tasks. However, since node classification only involves modeling node-level manifolds and is relatively simple, the performance improvement is less significant compared to graph regression tasks.

### 4.4 Ablation Study

As shown in Table 4, we further investigate the contribution of each component in GeoMancer. Notably, the substantial improvement in molecular validity primarily results from the self-guidance mechanism, which effectively exploits the complex geometry of the latent space to guide the generation process. In contrast, the increase in Novelty arises from the joint effect of self-guidance and manifold-constrained conditional generation. Additionally, the Riemannian model significantly enhances the model's ability to capture the underlying data distribution, showing better performance in FCD and NSPDK.

### 4.5 Visualization

To demonstrate the effectiveness of manifold selection, we visualize the decoupled manifolds and their weights on the ZINC12k dataset. As shown in Fig. 3, our approach captures geometric priors with diverse curvature representations, dynamically leveraging them across task levels. For example, at the graph level, molecular solubility is mainly influenced by hyperbolic and spherical features, with Euclidean features playing a smaller role. At the node and edge levels, each manifold contributes more evenly, with the visualization highlighting how different spaces adapt to the data's structural characteristics. These findings show that manifold selection enhances representational diversity and reveals how geometry impacts multi-level features.

## 5 Conclusion

In this work, we present GeoMancer, a Riemannian graph diffusion framework that unifies generation and prediction by explicitly modeling manifold signatures in graph data. By replacing unstable exponential mappings with a Riemannian gyrokernel and decoupling multi-level features across task-specific manifolds, our method mitigates numerical instability while preserving non-Euclidean geometric priors. Moreover, manifold-constrained diffusion and self-guided generation ensure that samples remain consistent with their underlying manifold distributions. Experiments on several datasets demonstrate the advantages of GeoMancer in classification, regression, and generation.

## Acknowledgement

The corresponding authors are Xingcheng Fu. This paper is supported by Beijing Natural Science Foundation (QY24129), and the National Natural Science Foundation of China (No.62462007 and No.62302023), Research Fund of Guangxi Key Lab of Multi-source Information Mining & Security (MIMS24-12), and The Basic Ability Enhancement Program for Young and Middle-aged Teachers of Guangxi (No.2024KY0073). We owe sincere thanks to all co-authors for their valuable efforts and contributions.

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

# A  Proof and Derivation

In this section ,we proof the Proposition 3.1.

**Definition** Given a Riemannian manifold $(M^n, g)$ and a function mapping $\varphi \in C^\infty(M)$, the differential map is linear for all $x \in M$. The gradient $\nabla_g \varphi$ of a function f is denoted as:

$$\langle \nabla_g \varphi(x_i), X_{x_i} \rangle_{g(x)} = d_{x_i} \varphi(X_{x_i}) \tag{12}$$

.

where $d_{x_i} \varphi : T_{x_i} M \to \mathbb{R}$ is linear for all $X_{x_i} \in T_{x_i} M$.

The divergence is the operator $\mathrm{div}_g : \Gamma_{C^\infty}(TM) \to C^\infty(M)$ making

$$d(\iota_X \omega_g) = \mathrm{div}_g X \cdot \omega_g \quad \text{for all} X \in \Gamma_{C^\infty}(TM). \tag{13}$$

where $\Gamma_{C^\infty}(TM)$ represents the space of smooth sections of the tangent bundle.

The Laplacian on $(M, g)$ is the operator defined as $\Delta_g = -\mathrm{div}_g \circ \nabla_g$.

**Lemma A.1.** *For any smooth functions $\varphi, \psi \in C^\infty(M)$, the Laplace-Beltrami operator $\Delta_g$ satisfies the following product rule:*

$$\Delta_g(\varphi \cdot \psi) = \psi \Delta_g \varphi + \varphi \Delta_g \psi - 2 \langle \nabla_g \varphi, \nabla_g \psi \rangle_g . \tag{14}$$

*This identity generalizes the classical product rule of the Laplacian to Riemannian manifolds, where$\langle \cdot, \cdot \rangle_g$ represents the Riemannian inner product with the metric g.*

*Proof.* To derive the formula for the divergence of a vector field on a Riemannian manifold, we begin by considering a smooth vector field

$$X = \sum_{j=1}^n b_j \frac{\partial}{\partial x_j} \in \Gamma_{C^\infty}(TM).$$

We express the contraction of $X$ with the volume form $\omega_g$:

$$\begin{aligned}
\iota_X \omega_g \left( \frac{\partial}{\partial x_1}, \ldots, \frac{\hat{\partial}}{\partial x_i}, \ldots, \frac{\partial}{\partial x_n} \right) &= \omega_g \left( X, \frac{\partial}{\partial x_1}, \ldots, \frac{\hat{\partial}}{\partial x_i}, \ldots, \frac{\partial}{\partial x_n} \right) \\
&= (-1)^{i-1} \omega_g \left( \frac{\partial}{\partial x_1}, \ldots, X, \ldots, \frac{\partial}{\partial x_n} \right) \\
&= (-1)^{i-1} \sqrt{|\det g|} dx_1 \wedge \cdots \wedge dx_n \left( \frac{\partial}{\partial x_1}, \ldots, X, \ldots, \frac{\partial}{\partial x_n} \right) \\
&= b_i (-1)^{i-1} \sqrt{|\det g|}.
\end{aligned} \tag{15}$$

Next, we compute the exterior derivative of the contraction:

$$\begin{aligned}
d(\iota_X \omega_g) &= d \left( \sum_{i=1}^n b_i (-1)^{i-1} \sqrt{|\det g|} dx_1 \wedge \cdots \wedge \hat{dx_i} \wedge \cdots \wedge dx_n \right) \\
&= \sum_{i=1}^n (-1)^{i-1} \frac{\partial}{\partial x_i} (b_i \sqrt{|\det g|}) dx_i \wedge dx_1 \wedge \cdots \wedge \hat{dx_i} \wedge \cdots \wedge dx_n \\
&= \sum_{i=1}^n \frac{\partial}{\partial x_i} (b_i \sqrt{|\det g|}) dx_1 \wedge \cdots \wedge dx_n \\
&= \frac{1}{\sqrt{|\det g|}} \sum_{i=1}^n \frac{\partial}{\partial x_i} (b_i \sqrt{|\det g|}) \cdot \omega_g.
\end{aligned} \tag{16}$$

Since the divergence of a vector field is defined by $\operatorname{div}_g X = d(\iota_X \omega_g)/\omega_g$, we obtain:

$$\operatorname{div}_g X = \frac{1}{\sqrt{|\det g|}} \sum_{i=1}^{n} \frac{\partial}{\partial x_i}(b_i \sqrt{|\det g|}). \tag{17}$$

Now, we generalize this formula to the case where the vector field is multiplied by a smooth function $\varphi$:

$$\operatorname{div}_g(\varphi X) = \varphi \operatorname{div}_g X + \langle \nabla_g \varphi, X \rangle_g. \tag{18}$$

This formula plays a crucial role in deriving the Laplace-Beltrami operator's product rule. Using the definition $\Delta_g = -\operatorname{div}_g \nabla_g$, we proceed to compute $\Delta_g(\varphi\psi)$:

$$
\begin{aligned}
\Delta_g(\varphi\psi) &= -\operatorname{div}_g\left(\nabla_g(\varphi\psi)\right) \\
&= -\operatorname{div}_g\left(\varphi\nabla_g\psi + \psi\nabla_g\varphi\right) \\
&= -\left[\operatorname{div}_g(\varphi\nabla_g\psi) + \operatorname{div}_g(\psi\nabla_g\varphi)\right] \\
&= -\left[\varphi\operatorname{div}_g(\nabla_g\psi) + \langle\nabla_g\varphi, \nabla_g\psi\rangle_g + \psi\operatorname{div}_g(\nabla_g\varphi) + \langle\nabla_g\psi, \nabla_g\varphi\rangle_g\right] \\
&= -\left[\varphi\Delta_g\psi + \psi\Delta_g\varphi + 2\langle\nabla_g\varphi, \nabla_g\psi\rangle_g\right] \\
&= \psi\Delta_g\varphi + \varphi\Delta_g\psi - 2\langle\nabla_g\varphi, \nabla_g\psi\rangle_g.
\end{aligned}
\tag{19}
$$

Thus, we have established the product rule for the Laplace-Beltrami operator on a Riemannian manifold.

$\square$

Here we begin to proof the proposition 3.1:

*Proof.* According to mathematical induction,

**Base Case: L=1**

For $L = 1$, the theorem reduces to the action of the Laplace-Beltrami operator on a single function $g_1$. By definition, if $g_1$ is an eigenfunction of $\Delta_{\mathcal{M}}$ with eigenvalue $\lambda_1$, we have:

$$\Delta_{\mathcal{M}} g_1 = \lambda_1 g_1, \tag{20}$$

which trivially satisfies the theorem. This establishes the base case.

**Inductive Hypothesis: L=N**

Assume the theorem holds for $L = N$, i,e.,for any product of $N$ eigenfunctions $g_1, g_2, \ldots, g_N$ with corresponding eigenvalues $\lambda_1, \lambda_2, \ldots, \lambda_N$, the Laplace-Beltrami operator acts as:

$$\Delta_{\mathcal{M}}\left(\prod_{i=1}^{N} g_i\right) = \left(\sum_{i=1}^{N} \lambda_i\right)\prod_{i=1}^{N} g_i. \tag{21}$$

**Inductive Step: L=N+1**

We now prove the theorem for $L = N + 1$, we can get:

$$\Delta_{\mathcal{M}}(\prod_{i=1}^{N+1} g_i) = \Delta_{\mathcal{M}}\left[(\prod_{i=1}^{N} g_i)g_{N+1}\right]. \tag{22}$$

Considering the Eq. (21) and Lemma A.1 , we can derive it as:

$$\Delta_{\mathcal{M}}(\prod_{i=1}^{N+1} g_i) = \Delta_{\mathcal{M}} \left[ \left( \prod_{i=1}^{N} g_i \right) g_{N+1} \right]$$

$$= g_{N+1} \Delta_{\mathcal{M}} \prod_{i=1}^{N} g_i + \prod_{i=1}^{N} g_i \Delta_{\mathcal{M}} g_{N+1} - 2 \left\langle \nabla_{\mathcal{M}} \prod_{i=1}^{N} g_i, \nabla_{\mathcal{M}} g_{N+1} \right\rangle_g$$

$$= g_{N+1} (\sum_{i=1}^{N} \lambda_i) \prod_{i=1}^{N} g_i + (\prod_{i=1}^{N} g_i) \lambda_{g_{N+1}} g_{N+1} \tag{23}$$

$$= (\sum_{i=1}^{N} \lambda_i) \prod_{i=1}^{N+1} g_i + \lambda_{N+1} \prod_{i=1}^{N+1} g_i$$

$$= (\sum_{i=1}^{N+1} \lambda_i) \prod_{i=1}^{N+1} g_i$$

It can be observed that the theorem holds for $L = N + 1$. Thus, we have successfully proven the proposition 3.1. □

## B  Preliminary

### B.1  Preliminary of Riemannian Geometry

In this section, we provide a detailed introduction to Riemannian geometry.

A smooth manifold $M$ is termed a Riemannian manifold when equipped with a Riemannian metric $g$. Curvature $c$ is a crucial measure that quantifies the extent of geodesic bending. For each point $x \in M$, there exists a tangent space $T_x M \subseteq \mathbb{R}^d$ that surrounds $x$, where the metric $g$ is applied to determine the manifold's shape. The relationship between the tangent space and the manifold is established through exponential and logarithmic maps. Specifically, the exponential map at point $x$, represented as $\exp_x^c(\cdot) : T_x M \to M$, transforms points from the tangent space into the manifold, while the logarithmic map $\log_x^c(\cdot) = (\exp_x^c(\cdot))^{-1}$ serves as its inverse..

In this paper, we use three geometric spaces of different curvature to form a product Riemannian manifold space: Euclidean space ($c = 0$), hyperbolic space ($c < 0$), and spherical space ($c > 0$).

**Hyperbolic space**. A hyperbolic space is defined as $\mathbb{H}_c^d = \{\mathbf{x}_p \in \mathbb{R}^{d+1} : \langle \mathbf{x}_p, \mathbf{x}_p \rangle_{\mathcal{L}} = 1/c\}$, where $d$ represents the dimension and the inner product is defined as $\langle \mathbf{x}, \mathbf{y} \rangle_{\mathcal{L}} = -x_1 y_1 + \sum_{j=2} x_j y_j$). In a hyperbolic space, The geodesic distance between the two points is:

$$d(x, y) = \frac{1}{\sqrt{-c}} \operatorname{arccosh} \left( c * \langle \mathbf{x}, \mathbf{y} \rangle_{\mathcal{L}} \right). \tag{24}$$

The exponential map in hyperbolic space is defined as:

$$exp_{x_p}^c(x) = \cosh \left( \sqrt{-c} ||\mathbf{x}|| \right) \mathbf{x}_p + \sinh \left( \sqrt{-c} ||\mathbf{x}|| \right) \frac{\mathbf{x}}{\sqrt{-c} ||\mathbf{x}||}. \tag{25}$$

**Sphere space**. Sphere space is defined as $\mathbb{S}_c^d = \{\mathbf{x}_p \in \mathbb{R}^{d+1} : \langle \mathbf{x}_p, \mathbf{x}_p \rangle_{\mathbb{S}} = 1/c\}$, where the inner product is the standard Euclidean inner product $\langle \mathbf{x}, \mathbf{y} \rangle_{\mathbb{S}} = \sum_{j=1}^{d+1} x_j y_j$. The geodesic distance between the two points is:

$$d(x, y) = \frac{1}{\sqrt{c}} \arccos \left( c \langle \mathbf{x}, \mathbf{y} \rangle_{\mathbb{S}} \right). \tag{26}$$

The exponential map in spherical space is defined as:

$$exp_{x_p}^c(x) = \cosh \left( \sqrt{c} ||\mathbf{x}|| \right) \mathbf{x}_p + \sinh \left( \sqrt{c} ||\mathbf{x}|| \right) \frac{\mathbf{x}}{\sqrt{-c} ||\mathbf{x}||}. \tag{27}$$

A product manifold is the Cartesian product $\mathcal{P} = \times_{i=1}^{n_\mathcal{P}} \mathcal{M}_{c_i}^{d_i}$, where $c_i$ and $d_i$ are the curvature and dimensionality of the manifold $\mathcal{M}_{K_i}^{c_i}$ respectively. If we restrict $\mathcal{P}$ to be composed of the Euclidean plane $\mathbb{E}_{c_E}^{d_E}$, hyperboloids $\mathbb{H}_{c_j^H}^{d_j^H}$ and hyperspheres $\mathbb{S}_{c_j^S}^{d_j^S}$ of constant curvature, we can represent an arbitrary product manifold of model spaces as such:

$$\mathcal{P} = \mathbb{E}_{c_E}^{d_E} \times \left( \underset{j=1}{\overset{n_\mathbb{H}}{\times}} \mathbb{H}_{c_j^\mathbb{H}}^{d_j^\mathbb{H}} \right) \times \left( \underset{k=1}{\overset{n_\mathbb{S}}{\times}} \mathbb{S}_{c_k^\mathbb{S}}^{d_k^\mathbb{S}} \right) \tag{28}$$

In Riemannian machine learning, each layer requires the conversion between exponential and logarithmic mappings. Although this process is conceptually natural, it is computationally complex and prone to instability. Notably, the equations associated with exponential mappings often lead to instability issues, such as values becoming excessively large or small, resulting in NaN (Not a Number) problems. Consequently, it is frequently necessary to meticulously identify appropriate hyperparameters to mitigate these issues.

## B.2 Kernel Method

**Theorem B.1.** *Bochner's theorem [57]: For any shift-invariant continuous kernel $k(\boldsymbol{x}, \boldsymbol{y}) = k(\boldsymbol{x} - \boldsymbol{y})$ defined on $\mathbb{R}^n$, if $p(\omega)$ is its Fourier transform and $\xi_\omega(\boldsymbol{x}) = \exp(i\langle \boldsymbol{\omega}, \boldsymbol{x} \rangle)$. then $k$ is positive definite if and only if $p \geq 0$. In this case if we sample $\omega$ according to the distribution proportional to $p(\omega)$, the kernel $k$ can be expressed as:*

$$k(\boldsymbol{x} - \boldsymbol{y}) = \int_{\mathbb{R}^n} p(\boldsymbol{\omega}) \exp(i\langle \boldsymbol{\omega}, \boldsymbol{x} - \boldsymbol{y} \rangle) d\boldsymbol{\omega} = k(\boldsymbol{0}) \cdot \mathbb{E}_{\boldsymbol{\omega} \sim p} \left[ \xi_{\boldsymbol{\omega}}(\boldsymbol{x}) \xi_{\boldsymbol{\omega}}(\boldsymbol{y})^* \right]. \tag{29}$$

Since both the probability distribution $p(\omega)$ and and the kernel $k$ are real, the integral is unchanged when we replace the exponential with a cosine. Leveraging this property, [29] developed a hyperbolic Laplacian feature function within hyperbolic space $\mathbb{H}_c^d$, yielding a hyperbolic Laplacian feature that approximates an invariance kernel in $\mathbb{H}_c^d$:

$$\text{HyLa}_{\lambda, b, \boldsymbol{\omega}}(\boldsymbol{z}) = \exp\left( \frac{n-1}{2} \langle \omega, \boldsymbol{z} \rangle_H \right) \cos\left( \lambda \langle \boldsymbol{\omega}, \boldsymbol{z} \rangle_H + b \right). \tag{30}$$

[18] generalized it to the more general Riemannian manifold. The Laplacian features of the Riemannian space are extracted by deriving the eigenfunction in the gyrovector ball $\mathbb{G}_\kappa^n$:

$$\text{gF}_{\boldsymbol{\omega}, b, \lambda}^\kappa(\boldsymbol{x}) = A_{\boldsymbol{\omega}, \boldsymbol{x}} \cos\left( \lambda \langle \boldsymbol{\omega}, \boldsymbol{x} \rangle_\kappa + b \right), \boldsymbol{x} \in \mathbb{G}_\kappa^n, \tag{31}$$

where $A_{\boldsymbol{\omega}, \boldsymbol{x}} = \exp\left( \frac{n-1}{2} \langle \omega, x \rangle_\kappa \right)$, $\langle \omega, x \rangle_\kappa = \log \frac{1 + \kappa \|\boldsymbol{x}\|^2}{\|\boldsymbol{x} - \boldsymbol{\omega}\|^2}$

Using the Eq. (31), a generalized Fourier map $\phi_{\text{gF}}(\boldsymbol{x})$ can be constructed to estimate an equidistant-invariant kernel on a Riemannian space. Moreover, this kernel can be seen as a generalization of Poisson's kernel in hyperbolic space.

This kernel method can be applied to two types of features: node embeddings and feature embeddings. For node embeddings, a Riemannian feature representation must first be constructed for each individual node $z_i$. Then, the inner product $\langle \phi_{\text{gF}}(zi), \phi\text{gF}(\boldsymbol{z}_j) \rangle$ between nodes $v_i$ and $v_j$ approximates a kernel function $k(z_i, z_j)$. The optimization of $\boldsymbol{z}_i$ is driven by the goal of learning an effective kernel over the product space to support downstream tasks. For feature embeddings, we represent normalized node feature as $\mathbf{X} \in \mathbb{R}^{n \times d}$ and inital Riemannian embedding as $z_i$. Then it can be calculated by $\sum_{k=1}^d \mathbf{X}_{ik} \phi_{\text{gF}}(\boldsymbol{z}_k)$. Its inner product between two features is:

$$\langle \sum_{k=1}^d \mathbf{X}_{ik} \phi_{\text{gF}}(\boldsymbol{z}_k), \sum_{l=1}^d \mathbf{X}_{jl} \phi_{\text{gF}}(\boldsymbol{z}_l) \rangle = \sum_{k,l=1}^d \mathbf{X}_{ik} \mathbf{X}_{jl} \langle \phi_{\text{gF}}(\boldsymbol{z}_k), \phi_{\text{gF}}(\boldsymbol{z}_l) \rangle. \tag{32}$$

## B.3 Preliminary of Diffusion Model

Denoising Diffusion Probabilistic Models (DDPMs) [13] are a class of generative models based on diffusion processes that generate high-quality samples by simulating a step-by-step denoising process, transforming noise into realistic data. The core idea of DDPMs is to construct a generative model through two complementary processes: a forward process that gradually adds noise to data, and a reverse process that learns to iteratively denoise and reconstruct the data distribution.

**Forward Process**: the forward process $q(z|x)$ is the variance-preserving Markov process:

$$q(\mathbf{z}_\lambda|\mathbf{x}) = \mathcal{N}(\alpha_\lambda \mathbf{x}, \sigma_\lambda^2 \mathbf{I}), \text{where} \alpha_\lambda^2 = 1/(1 + e^{-\lambda}), \sigma_\lambda^2 = 1 - \alpha_\lambda^2 \tag{33}$$

For intermediate steps, the transition between noise levels is given by:

$$q(\mathbf{z}_\lambda|\mathbf{z}_{\lambda'}) = \mathcal{N}((\alpha_\lambda/\alpha_{\lambda'})\mathbf{z}_{\lambda'}, \sigma_{\lambda|\lambda'}^2 \mathbf{I}), \text{where} \lambda < \lambda', \sigma_{\lambda|\lambda'}^2 = (1 - e^{\lambda-\lambda'})\sigma_\lambda^2 \tag{34}$$

**Reverse Process**: The reverse process is a generative model that starts from a prior distribution $p_\theta(\mathbf{z}_{\lambda_{\min}}) = \mathcal{N}(\mathbf{0}, \mathbf{I})$ and reconstructs the real data distribution by iteratively denoising the data. The reverse transition is modeled as:

$$p_\theta(\mathbf{z}_{\lambda'}|\mathbf{z}_\lambda) = \mathcal{N}(\tilde{\boldsymbol{\mu}}_{\lambda'|\lambda}(\mathbf{z}_\lambda, \mathbf{x}_\theta(\mathbf{z}_\lambda)), (\tilde{\sigma}_{\lambda'|\lambda}^2)^{1-v}(\sigma_{\lambda|\lambda'}^2)^v) \tag{35}$$

where $\tilde{\boldsymbol{\mu}}_{\lambda'|\lambda}(\mathbf{z}_\lambda, \mathbf{x}) = e^{\lambda-\lambda'}(\alpha_{\lambda'}/\alpha_\lambda)\mathbf{z}_\lambda + (1 - e^{\lambda-\lambda'})\alpha_{\lambda'}\mathbf{x}$, and $\tilde{\sigma}_{\lambda'|\lambda}^2 = (1 - e^{\lambda-\lambda'})\sigma_{\lambda'}^2$.

The model is trained to predict the noise at each step by minimizing the objective function, which measures the discrepancy between the predicted noise and the actual noise added during the forward process. Specifically, the training objective is formulated as: $\mathbb{E}_{\boldsymbol{\epsilon},\lambda}\left[\|\boldsymbol{\epsilon}_\theta(\mathbf{z}_\lambda) - \boldsymbol{\epsilon}\|_2^2\right]$

To enhance the controllability and quality of generated samples, DDPMs often employ guidance mechanisms:

**Classifier Guidance**: Classifier guidance [58] introduces a conditional diffusion process by leveraging a pre-trained classifier $p_\theta(\mathbf{c}|\mathbf{z}_\lambda)$. The guided noise prediction is given by:

$$\tilde{\epsilon}_\theta(\mathbf{z}_\lambda, \mathbf{c}) = \epsilon_\theta(\mathbf{z}_\lambda, \mathbf{c}) - w\sigma_\lambda \nabla_{\mathbf{z}_\lambda} \log p_\theta(\mathbf{c}|\mathbf{z}_\lambda) \approx -\sigma_\lambda \nabla_{\mathbf{z}_\lambda}[\log p(\mathbf{z}_\lambda|\mathbf{c}) + w \log p_\theta(\mathbf{c}|\mathbf{z}_\lambda)], \quad (36)$$

where $w$ controls the strength of the guidance. This can be interpreted as:

$$\tilde{\epsilon}_\theta(\mathbf{z}_\lambda, \mathbf{c}) \approx -\sigma_\lambda \nabla_{\mathbf{z}_\lambda}\left[\log p(\mathbf{z}_\lambda|\mathbf{c}) + w \log p_\theta(\mathbf{c}|\mathbf{z}_\lambda)\right]. \tag{37}$$

**Classifier-free Guidance**: Classifier-free guidance [59] eliminates the need for a separate classifier by jointly training conditional and unconditional models. The guided noise prediction is computed as:

$$\tilde{\epsilon}_\theta(\mathbf{z}_\lambda, \mathbf{c}) = (1 + w)\epsilon_\theta(\mathbf{z}_\lambda, \mathbf{c}) - w\epsilon_\theta(\mathbf{z}_\lambda) \tag{38}$$

where $w$ is a hyperparameter that controls the strength of conditional guidance. This approach simplifies the training process while maintaining high controllability.

**Manifold-Constrained Classifier-free Guidance**: [20] model condition generation as solving an inverse problem:

$$\min_{\boldsymbol{x}\in\mathcal{M}} \ell_{sds}(x), \quad \ell_{sds}(\boldsymbol{x}) := \|\boldsymbol{\epsilon}_\theta(\sqrt{\bar{\alpha}_t}\boldsymbol{x} + \sqrt{1 - \bar{\alpha}_t}\boldsymbol{\epsilon}, \boldsymbol{c}) - \boldsymbol{\epsilon}\|_2^2 \tag{39}$$

This implies that the goal is to identify solutions on the clean manifold $\mathcal{M}$ that optimally aligns with the condition $c$. The resulting sampling process from reverse diffusion is then given by

$$x_{t-1} = \sqrt{\bar{\alpha}_{t-1}}\left(\hat{x}_\varnothing - \gamma_t \nabla_{\hat{x}_\varnothing}\ell_{sds}(\hat{x}_\varnothing)\right) + \sqrt{1 - \bar{\alpha}_{t-1}}\hat{\epsilon}_\varnothing. \tag{40}$$

Table 5: Overview of the datasets and metrics used in the paper.

| Dataset | #Graphs | Avg. #nodes | Avg. #edges | Prediction Level | Task | Metric |
|---------|---------|-------------|-------------|------------------|------|--------|
| QM9 | 130,000 | 18.0 | 37.3 | graph | regression | Mean Absolute Error |
| ZINC | 12,000 | 23.2 | 24.9 | graph | regression | Mean Absolute Error |
| Cora | 1 | 2708 | 10,556 | node | 7-way classification | Accuracy |
| PubMed | 1 | 19717 | 88648 | node | 3-way classification | Accuracy |
| Photo | 1 | 7650 | 238,162 | node | 8-way classification | Accuracy |
| Physics | 1 | 34,493 | 495,924 | node | 5-way classification | Accuracy |

we can equivalently write the loss as $\ell_{sds}(\boldsymbol{x}) = \frac{\bar{\alpha}_t}{1-\bar{\alpha}_t}\|\boldsymbol{x} - \hat{\boldsymbol{x}}_c\|^2$, so it can be written as:

$$\boldsymbol{x}_{t-1} = \sqrt{\bar{\alpha}_{t-1}}\left(\hat{x}_\varnothing + \lambda(\hat{x}_{\boldsymbol{c}} - \hat{x}_\varnothing)\right) + \sqrt{1 - \bar{\alpha}_{t-1}}\hat{\boldsymbol{\epsilon}}_\varnothing \tag{41}$$

## C  Experiment details

**Diffusion Process**. In all experiments, we employ a diffusion process with $T = 1000$ diffusion steps, diffusion steps, parameterized by a linear schedule for $\alpha_t$ and a corresponding decay for $\bar{\alpha}_t$. For inference, we adopt the DDPM framework.

**Model Architechture**. For the node-level task, including all node classification datasets, we use an MPNN-centric model as the encoder, which consists of GCN, GIN and GAT. Typically, we use a 5-layer architecture with residual connections and normalization layers to facilitate optimization. The optimizer is AdamW, with a learning rate of 1e-3 and a weight decay of 1e-5. The dimension of final hidden layer is set to 4. Since the node classification task only considers the node features, we only study the reconstruction loss and Riemann decoupling of the node features. Here, we initialize the curvature of a product manifold consisting of three gyroscopic space vectors with curvature -1,0,1, respectively, with dimension 4. The random Laplacian map is then calculated separately and the underlying spatial features are transformed into Riemannian Spaces. Finally, they are weighted by a linear layer and then decoded by a linear layer.

For graph-level tasks, such as graph generation and regression, we employ edge-enhanced graph transformers as the backbone network, incorporating position embeddings to capture structural information. For the decoder, we design a Riemannian decoupling layer for each subtask, including node-level reconstruction, edge-level reconstruction, and graph-level property reconstruction. After the Riemannian decoupling layers, a linear layer aggregates representations from each Riemannian space, and the final results are produced through an additional linear layer. The optimizer is AdamW, with a learning rate of 1e-4 and a weight decay of 1e-6. The dimension of final hidden layer is set to 16 or 32.

## D  Experiment Analysis

### D.1  Analysis of Riemannian Autoencoder

To better analyze the impact of our Riemannian autoencoder, we conduct a detailed evaluation on the ZINC12 graph regression task. Figure D.1 reports the convergence speed of MAE during model training. We compare our approach with the model without Riemannian block. The learning rate setting of the optimizer remains consistent, all being 1e-5. The results show that the convergence speed of this model is significantly accelerated. Furthermore, our final convergence result is superior to removing the Riemannian block. We attribute this improvement to the Riemannian decoupler, which effectively decouples each feature onto the appropriate product manifold, thereby promoting more efficient learning.

## E  Limitation

Due to the limitation of computing resources, our model is not large enough and lacks the verification of the scaling law of the diffusion method on graph tasks.

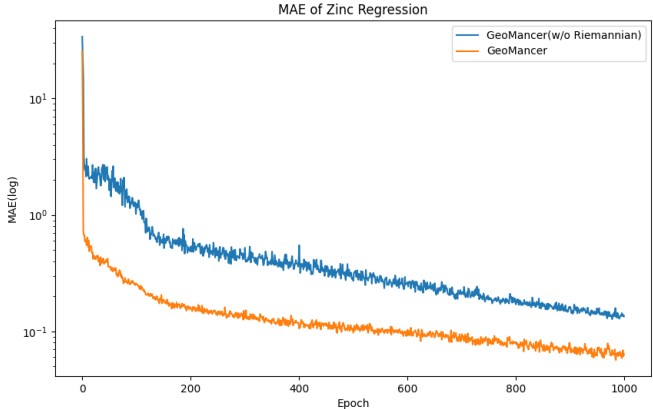

Figure 5: MAE during the training in graph regression.

# F Broader Impact

This paper aims to advance the field of graph generation technologies. Our work contributes to the field of Machine Learning and has many potential societal consequences. It may play an important role in understanding fields such as drug generation and recommendation systems based on graph structures from a geometric perspective. However, we believe that there are no negative impacts that need clarification.

