# OpenReview forum: "Toward a Unified Geometry Understanding : Riemannian Diffusion Framework for Graph Generation and Prediction"
_NeurIPS.cc/2025/Conference — NeurIPS 2025 poster_

### Official Review · Reviewer_8zMr · 2025-06-23

**Clarity:** 3
**Significance:** 3
**Originality:** 3
**Rating:** 4
**Confidence:** 1

**Summary:**

The paper introduces GeoMancer, a framework for graph generation and prediction using Riemannian diffusion. This framework explicitly models various geometric curvatures within distinct task-specific manifolds. To address numerical instability during the encoding process, the authors use an isometry-invariant Riemannian "gyrokernel" instead of the traditional exponential map. To avoid manifold drift during generation, they implement a manifold-constrained diffusion process along with a self-guided strategy for unconditional sampling. The experiments conducted on multiple benchmark datasets demonstrate advancements over both Euclidean and previous hyperbolic graph generation methods.

**Questions:**

What is the computational cost of computing the gyrokernel?

**Ethical Concerns:**

["NO or VERY MINOR ethics concerns only"]

**Final Justification:**

I am not an expert in this topic, but to the best of my knowledge, I think the authors did a good job in the paper.

**Limitations:**

Yes

**Paper Formatting Concerns:**

I think that some of the citations should be improved. Conference names should be spelled out in full in the references: use “Advances in Neural Information Processing Systems” instead of “NeurIPS”, and “International Conference on Learning Representations” instead of “ICLR”. This can be done by exporting the Bibtex from google scholar.

**Quality:**

3

**Strengths And Weaknesses:**

I am not very familiar with ideas of diffusion, but I have done my best to identify the strengths and weaknesses of the paper. I hope this might help the authors improve their paper slightly.

**Strengths**

- The motivation of the paper is clear, and it is to be followed by good empirical results.
- I found the use of the Riemannian gyrokernel in place of the exponential map an interesting approach to address numerical instability, which deviates from the traditional approach in the literature.
- The results in the paper seem to consistently improve multiple graph-generation and prediction benchmarks.

**Weaknesses**

- The authors could potentially add some additional benchmarks [1] to increase the validity of their results.

[1] Qin, Yiming, et al. "Defog: Discrete flow matching for graph generation." arXiv preprint arXiv:2410.04263 (2024).

---

> ### Author Rebuttal · Authors · 2025-07-29
>
> Dear Reviewer 8zMr,
>
> We sincerely thank the reviewer for the detailed comments and insightful questions. Our response to your comments one by one as follows.
>
> ---
>
> > W1: The authors could potentially add some additional benchmarks [1] to increase the validity of their results.
> >
>
> Reply to W1: Thank you for pointing out the model and benchmarks we had previously overlooked. We will cite this work in the revised version and try to add the relevant metrics to the MOSES dataset, and incorporate the baseline model.
>
> ---
>
> > Q1: What is the computational cost of computing the gyrokernel?
> >
>
> Reply to Q1: Here, we analyze the complexity of our Riemannian gyrokernel mappings from theory and experiment.
>
> Theory: Suppose that a general Fourier mapping $\phi_{\mathrm{gF}}(x)$ consists of m eigenfunctions and the number of representation vectors $V$ in different Riemannian spaces is $n$. First, the computation of each eigenfunction $\mathrm{gF}_{\boldsymbol{\omega},b,\lambda}^{\kappa}(\boldsymbol{x})$ involves only simple operations with the time complexity of $O(1)$. Thus ,the whole process of calculating $\overline{V}$ is O(m*n).  It is worth noting that since the settings of m and n are very small, typically m*n<10. the process is therefore very fast.
>
> Experiment: We report the time consumption in Table 1. Experimental results show that our mapping remains almost identical to the original method without significant time loss.
>
> Table 1 time consumption on photo dataset in one epoch
>
> | Method | Time(s) |
> | --- | --- |
> | GeoMancer(w/o riemannian） | 1.5 |
> | GeoMancer | 1.6 |

---

> > ### Comment · Reviewer_8zMr · 2025-08-04
> > **Reply**
> >
> > Thank you for addressing my comments. As I mentioned, I am not too familiar with the topic at hand, but from what I can see, the authors did a good job in the submission and in the rebuttal.

---

> > > ### Author Response · Authors · 2025-08-07
> > > **Reply**
> > >
> > > We sincerely appreciate your feedback and are delighted to address your concerns. Thank you for your kind recognition and positive evaluation of our work.

---

### Official Review · Reviewer_aMiG · 2025-06-29

**Clarity:** 3
**Significance:** 4
**Originality:** 4
**Rating:** 5
**Confidence:** 5

**Summary:**

This work introduces a geometry-based diffusion mechanism that adapts to the intrinsic curvature of multi-level graph features, from node, edge and graph feature. It first embeds multi-level graph data into a product manifold using a specially designed gyroscopic kernel.

**Questions:**

What is the denoising process like under the self-guided mechanism? Can the author provide visualization results? Why not use other Riemannian diffusion models? Is there any relationship between the pseudo-labels generated by molecules and their chemical properties? Can the author calculate the jaccard coefficients of the two categories after clustering based on their chemical properties?

**Ethical Concerns:**

["NO or VERY MINOR ethics concerns only"]

**Final Justification:**

I have carefully reviewed the authors’ rebuttal, and all of my previous questions have been clearly addressed. Thus, I will maintain my score.

**Limitations:**

yes

**Quality:**

4

**Strengths And Weaknesses:**

Strength:
S1: The motivation behind this work is clearly articulated, with well-justified explanations for the design of each component and the specific challenges they address.
S2: The article is grounded in solid theoretical foundations, offering rigorous proofs on the construction and decoupling of product manifolds through the use of gyroscopic kernels.
S3: The self-guidance mechanism based on geometric features is particularly well-conceived. Experimental results suggest it plays a significant role in enhancing generative performance.
S4: The experimental evaluation is comprehensive, with the model’s effectiveness validated across multiple tasks. The ablation studies convincingly demonstrate the contribution of each individual component.
Weakness:
W1: The lack of visual representation of the generation process makes it impossible to vividly demonstrate how this self-guiding mechanism is generated.
W2: The paper lacks a comparative discussion of existing Riemannian diffusion approaches[1][2]. It is important to explain why alternative Riemannian diffusion models were not adopted or compared.
W3: The self-guiding mechanism warrants a deeper analysis.  For instance, it remains unclear whether the generated labels reflect underlying chemical properties of the molecules and, if so, how such relationships are captured or influenced by the guidance strategy.
[1] Generative Modeling on Manifolds Through Mixture of Riemannian Diffusion Processes.
[2] Riemannian Diffusion Models.

---

> ### Author Rebuttal · Authors · 2025-07-29
>
> Dear Reviewer aMiG,
>
> We sincerely thank the reviewer for the detailed comments and insightful questions. Our response to your comments one by one as follows.
>
> > W1: The lack of visual representation of the generation process makes it impossible to vividly demonstrate how this self-guiding mechanism is generated. Q1: What is the denoising process like under the self-guided mechanism?
> >
>
> Reply to W1: We are very sorry that due to the latest rebottal notification, we are unable to provide the relevant figures. The self-guiding mechanism enables the model to generate features that are more similar to the original representation. In the later stage of denoising, the features of the same label will be closer. However, if the self-guiding mechanism is removed, the generated features will be scattered at various locations. We will add relevant visualization content in the revised version.
>
> ---
>
> > W2: The paper lacks a comparative discussion of existing Riemannian diffusion approaches[1][2]. It is important to explain why alternative Riemannian diffusion models were not adopted or compared. Q2: Why not use other Riemannian diffusion models?
> >
>
> Reply to W2: We clarify that one of the benefits for this design is to consider the efficiency issue of implementing diffusion directly on Riemannian geometry. **Benefiting from the fact that the feature remains in the Euclidean space after the generalized Fourier mapping, this design does not require a complex execution of the diffusion process on the Riemannian space no matter whether to use Riemannian autoencoder.** In addition, the Riemannian kernel method can indirectly enhance the representation of the latent space and thus improve the performance of diffusion model. This flexibility allows our architecture to be compatible with pre-trained graph latent diffusion models in the future without training a specialized Riemannian diffusion model from scratch.
>
> ---
>
> > W3:  The self-guiding mechanism warrants a deeper analysis. For instance, it remains unclear whether the generated labels reflect underlying chemical properties of the molecules and, if so, how such relationships are captured or influenced by the guidance strategy. Q3: Can the author calculate the jaccard coefficients of the two categories after clustering based on their chemical properties?
> >
>
> Reply to W3: This is indeed a very valuable question. We conducted an analysis on the QM9 dataset to examine the relationship between clustering labels and the chemical attribute $\mu$. Since the chemical attribute values are continuous numerical data, we first sorted them based on their magnitudes and divided them into corresponding sets. We then computed the Jaccard index between these sets and those derived from the clustering labels. **The average Jaccard score obtained was 0.1286, indicating that the self-guiding mechanism is only weakly related to the underlying chemical properties.**
>
> Although there exists a certain correlation between molecular structure and chemical properties, the representations used for generation may exhibit weaker correlations with chemical attributes, as they focus on different levels of information. We believe that the pseudo-labels produced by the self-guiding mechanism are primarily influenced by structural topology, which include the reconstructed representation information of nodes and edges.

---

> > ### Comment · Reviewer_aMiG · 2025-08-06
> >
> > I have carefully reviewed the authors’ rebuttal, and all of my previous questions have been clearly addressed.

---

> > > ### Author Response · Authors · 2025-08-07
> > > **Reply**
> > >
> > > We sincerely appreciate your feedback and are delighted that  your concerns have been solved. Thank you for your positive evaluation of our work.

---

### Official Review · Reviewer_c5kY · 2025-06-30

**Clarity:** 3
**Significance:** 3
**Originality:** 4
**Rating:** 5
**Confidence:** 4

**Summary:**

This paper proposed a method GeoMancer which approaches graph-structured data from a fundamentally different angle by modeling multi-level latent geometry in curved space.

**Questions:**

I am not clear about the ability of the diffusion model to handle heterogeneous manifolds. Therefore, I hope the author can discuss the following issues with me in detail. I will consider improving my score.

Q1: In Fig1(b),in what aspects is the complexity of multi-level data manifolds reflected? Are their curvatures the same? For example, for node classification or link prediction tasks, are node features and edge features completely different?

Q2: Can the diffusion model effectively capture data manifolds? Are there any previous studies or theoretical derivations?

Q3: What is the computational complexity of the gyroscope kernel method mentioned in the paper?

**Ethical Concerns:**

["NO or VERY MINOR ethics concerns only"]

**Final Justification:**

Embedding graph geometry through Riemannian geometry and curvature computation is highly challenging and interesting. The authors have done an excellent job. After reading other reviews, in my opinion, all major issues have been well addressed. Therefore, I keep my score at 5.

**Limitations:**

Yes.

**Paper Formatting Concerns:**

NA.

**Quality:**

4

**Strengths And Weaknesses:**

**Strength**
1. The paper introduces a novel approach by leveraging Riemannian geometry to model the feature space of multi-level graph data, offering a novel perspective on geometric representation in graph learning.
2. The integration of a self-guided mechanism with manifold-constrained diffusion is particularly innovative. This design effectively utilizes both explicit and implicit geometric cues to enhance representation quality.
3. The visualization experiments are both informative and aesthetically presented, clearly demonstrating the model’s ability to capture and interpret geometric structures.

**Weakness**
1. The explanation of Figure 1(b) could be expanded. It remains unclear whether different feature levels correspond to distinct curvatures, dimensions, or structural complexities, and how these aspects interact.
2. I'm not very familiar with the ability of diffusion to capture data manifolds. The paper lacks a thorough discussion on how well the diffusion process captures data manifolds, particularly in the context of modeling multiple, potentially heterogeneous, geometric levels.
3. The computational cost of the proposed method is not addressed.  A discussion of its time complexity would help assess the scalability and practicality of the approach.

---

> ### Author Rebuttal · Authors · 2025-07-29
>
> Dear Reviewer c5kY,
>
> We sincerely thank the reviewer for the detailed comments and insightful questions. Our response to your comments one by one as follows.
>
> ---
>
> > W1: The explanation of Figure 1(b) could be expanded. It remains unclear whether different feature levels correspond to distinct curvatures, dimensions, or structural complexities, and how these aspects interact. Q1: In Fig1(b),in what aspects is the complexity of multi-level data manifolds reflected? Are their curvatures the same? For example, for node classification or link prediction tasks, are node features and edge features completely different?
> >
>
> Reply to W1: On the one hand, features at different levels exhibit varying curvatures. We calculated the ricci curvature of different features as reported in Table 1. This indicates that the spatial curvatures of features across levels are distinct, highlighting the complexity of the data manifold. On the other hand, from the task perspective, different tasks encourage  different manifolds. Node classification encourages nodes with the same label to learn similar features, whereas link prediction primarily emphasizes structural features. Thus, the differing task objectives further contribute to the complexity of the data manifold.
>
> Table 1 curvature of multi-level embeddings
>
> | Node-level | Edge-level | Graph-level |
> | --- | --- | --- |
> | 0.3 | -0.6 | 0.1 |
>
> ---
>
> > W2:  I'm not very familiar with the ability of diffusion to capture data manifolds. The paper lacks a thorough discussion on how well the diffusion process captures data manifolds, particularly in the context of modeling multiple, potentially heterogeneous, geometric levels. Q2: Can the diffusion model effectively capture data manifolds? Are there any previous studies or theoretical derivations?
> >
>
> Reply to W2: To our knowledge, many studies[1][2][3] have demonstrated that diffusion models can effectively capture data manifolds. **The key insight is that the score function learned by a diffusion model tends to point in the normal direction towards the data manifold.** This property has been further leveraged to investigate intrinsic dimensions and other geometric characteristics of data manifolds.
>
> [1] Diffusion Models Encode the Intrinsic Dimension of Data Manifolds.
>
> [2] Adaptivity of Diffusion Models to Manifold Structures.
>
> [3] Convergence of Diffusion Models Under the Manifold Hypothesis in High‑Dimensions.
>
> ---
>
> > W3: The computational cost of the proposed method is not addressed. A discussion of its time complexity would help assess the scalability and practicality of the approach. Q3: What is the computational complexity of the gyroscope kernel method mentioned in the paper?
> >
>
> Reply to W3: Here, we analyze the complexity of our Riemannian mappings from theory and experiment.
>
> Theory: Suppose that a general Fourier mapping $\phi_{\mathrm{gF}}(x)$ consists of m eigenfunctions and the number of representation vectors $V$ in different Riemannian spaces is $n$. First, the computation of each eigenfunction $\mathrm{gF}_{\boldsymbol{\omega},b,\lambda}^{\kappa}(\boldsymbol{x})$ involves only simple operations with the time complexity of $O(1)$. Thus ,the whole process of calculating $\overline{V}$ is O(m*n).  It is worth noting that since the settings of m and n are very small, typically m*n<10. the process is therefore very fast.
>
> Experiment: We report the time consumption in Table 1. Experimental results show that our mapping remains almost identical to the original method without significant time loss.
>
> Table 1 time consumption on photo dataset in one epoch
>
> | Method | Time(s) |
> | --- | --- |
> | GeoMancer(w/o riemannian) | 1.5 |
> |GeoMancer | 1.6 |

---

> > ### Comment · Reviewer_c5kY · 2025-08-08
> >
> > Thanks for your responses, which have well addressed my concerns. I would like to keep the score at 5.

---

### Official Review · Reviewer_bYtP · 2025-07-01

**Clarity:** 3
**Significance:** 4
**Originality:** 3
**Rating:** 5
**Confidence:** 3

**Summary:**

Graph diffusion models offer a unified framework for tackling both graph generation and prediction tasks. To account for the non-Euclidean nature of graph-structured data, this paper proposes GeoMancer, a Riemannian diffusion approach that captures the intrinsic geometric nature of multi-level features. GeoMancer integrates Riemannian kernel autoencoders with manifold-constrained diffusion processes.  Experiments across multiple tasks demonstrate that GeoMancer achieves strong performance in both generation and prediction tasks.

**Questions:**

1. Could the authors provide more introductions and analyses about this kernel method?
2. How are the dimensions and manifolds selected?

**Ethical Concerns:**

["NO or VERY MINOR ethics concerns only"]

**Final Justification:**

I have read the rebuttal as well as the comments from other reviewers and found no major flaws. Concerns regarding the background introduction and learnable curvature are well addressed.  Meanwhile, I suggest the author incorporate the several revisions for their manuscript.

**Limitations:**

yes

**Quality:**

3

**Strengths And Weaknesses:**

## Strength:

1. The motivation of this work is well-grounded. While prior researches[1] on non-Euclidean graph data have primarily focused on modeling the geometric structure of individual features, this paper takes a significant step forward by exploring the complex geometry spaces of multi-level features.
2. The Riemann gyroscope kernel method is interesting. As the authors point out, the traditional exponential map often suffers from instability during optimization and requires meticulous parameter tuning. The gyroscope-based alternative offers a promising and potentially more robust replacement.
3. The paper provides solid theoretical grounding, including detailed derivations that justify the use of the Riemannian gyroscope kernel to model geometric structures over product manifolds.

## Weakness:
1. There is a lack of a more detailed introduction to the Riemannian kernel method, such as how much its distortion is and the influence of the number of basis functions on the mapping.
2. The author of the manifold signature only considered setting the learnable curvature to obtain it, but did not make a more detailed design for the selection of dimensions and manifolds.
3. In order for readers to better understand the background of geometric deep learning[1], more preliminary about Riemannian geometric methods should be added in the article and citations of relevant literature should be supplemented like [2-4] .

---

[1] Geometric deep learning: going beyond Euclidean data.

[2] Neural embeddings of graphs in hyperbolic space

[3] Computationally tractable riemannian manifolds for graph embeddings.

[4] Mixed-curvature Variational Autoencoders.

---

> ### Author Rebuttal · Authors · 2025-07-29
>
> Dear Reviewer bYtP,
>
> We sincerely thank the reviewer for the detailed comments and insightful questions. Our response to your comments one by one as follows.
>
> > W1: There is a lack of a more detailed introduction to the Riemannian kernel method, such as how much its distortion is and the influence of the number of basis functions on the mapping.
> Q1. Could the authors provide more introductions and analyses about this kernel method?
> >
>
> Reply to W1&Q1: There are many studies that use kernel methods or Laplace operators to analyze data manifold learning [1][2][3][4]. **This is an interesting mapping method that it essentially preserves the isometric invariance between two points while the feature still remain in Euclidean space.** [1] showed that the Laplace eigenfunctions can approximate the estimation of the kernel and replace the exponential mapping in hyperbolic space. Although it is not necessarily mathematically elegant enough, it is a very important property for model training, which means that we can simplify many additional Riemannian operators.
>
> [1] Random Laplacian Features for Learning with hyperbolic space. ICLR 2023
>
> [2] Motif-aware Riemannian Graph Neural Network with Generative-Contrastive Learning. AAAI 2024
>
> [3] Analysis on Manifolds via the Laplacian.
>
> [4] Analysis of the Laplacian on the complete Riemannian manifold.
>
> ---
>
> > W2: The author of the manifold signature only considered setting the learnable curvature to obtain it, but did not make a more detailed design for the selection of dimensions and manifolds.
> Q2: How are the dimensions and manifolds selected?
> >
>
> Reply to W2: As we explained in Section 3.2, for the learnable curvature, we adopt the gyrovector ball as the underlying manifold. Regarding the dimensionality, we currently lack an effective way to optimize it during training and can only set it as a fixed hyperparameter in advance. It is such a valuable comment worth exploring in the future.
>
> ---
>
> > W3: In order for readers to better understand the background of geometric deep learning[1], more preliminary about Riemannian geometric methods should be added in the article and citations of relevant literature should be supplemented like [2-4] .
> >
>
> Reply to W3: We sincerely appreciate the reviewer’s valuable suggestions.  These recommendations are instrumental in strengthening the background of our work. We will carefully incorporate this feedback by expanding the background discussion and adding relevant citations in the revised version.

---

> > ### Comment · Reviewer_bYtP · 2025-08-05
> >
> > Thank you for the response. I have read the rebuttal as well as the comments from other reviewers and found no major flaws. I have decided to raise my score for stronger support. Meanwhile, I suggest the author incorporate the following revisions for their manuscript:
> > - more introduction on Riemannian kernel method
> > - computational complexity (as in response to reviewer c5ky)
> > - discussions on the selection of dimensionality.

---

> > > ### Author Response · Authors · 2025-08-07
> > > **Reply**
> > >
> > > We are extremely grateful for your recognition of our work. We will add these contents in the revised version based on your suggestions.

---

### Decision · Program_Chairs · 2025-09-17

**Decision:**

Accept (poster)

**Comment:**

This paper introduces GeoMancer, a Riemannian diffusion framework for graph generation and prediction. The main contributions are: (1) a gyroscope-based Riemannian kernel that mitigates numerical instability compared to the exponential map, (2) a manifold-constrained diffusion process with a self-guided mechanism that ensures generated graphs remain aligned with manifold structures, and (3) comprehensive empirical results showing improvements over Euclidean and hyperbolic baselines across multiple tasks.  Reviewers highlighted the novelty of modeling multi-level geometric structures, the sound theoretical foundation, and the strong empirical validation. Notably, the unified framework proposed by this work closes the gap between two important problems via an interesting perspective.

For the camera-ready version, I encourage the authors to (1) expand the introduction and related work on Riemannian kernel methods and geometric deep learning (see the suggested literature by Reviewer bYtP), (2) provide more discussion and visualization of the self-guided mechanism and its implications for interpretability, and (3) include complexity analysis to further strengthen the practical impact of the framework.